# The Effect of the COVID-19 Pandemic on Self-Reported Health Status and Smoking and Drinking Habits in the General Urban Population

**DOI:** 10.3390/jcm12196241

**Published:** 2023-09-27

**Authors:** Magdalena Chlabicz, Aleksandra Szum-Jakubowska, Paweł Sowa, Małgorzata Chlabicz, Sebastian Sołomacha, Łukasz Kiszkiel, Łukasz Minarowski, Katarzyna Guziejko, Piotr P. Laskowski, Anna M. Moniuszko-Malinowska, Karol A. Kamiński

**Affiliations:** 1Department of Population Medicine and Lifestyle Diseases Prevention, Medical University of Bialystok, 15-269 Bialystok, Poland; chlabicz.m@gmail.com (M.C.); aleksandra.szum-jakubowska@umb.edu.pl (A.S.-J.); mailtosowa@gmail.com (P.S.); mchlabicz@op.pl (M.C.); sebastian.solomacha@sd.umb.edu.pl (S.S.); 2Department of Invasive Cardiology, Medical University of Bialystok, 15-276 Bialystok, Poland; 3Society and Cognition Unit, Institute of Sociology, University of Bialystok, 15-420 Bialystok, Poland; lukaszkiszkiel@gmail.com (Ł.K.); p.laskowski@uwb.edu (P.P.L.); 42nd Department of Lung Diseases and Tuberculosis, Medical University of Bialystok, Zurawia 14, 15-540 Bialystok, Poland; lukasz.minarowski@umb.edu.pl (Ł.M.); kguziejko@wp.pl (K.G.); 5Department of Infectious Diseases and Neuroinfection, Medical University of Bialystok, 15-540 Bialystok, Poland; annamoniuszko@op.pl; 6Department of Cardiology, Medical University of Bialystok, 15-276 Bialystok, Poland

**Keywords:** COVID-19, smoking habits, alcohol habits, health status

## Abstract

The coronavirus disease 2019 pandemic created a significant crisis in global health. The aim of the study was to compare the impact of the COVID-19 pandemic on self-rated health status and smoking and alcohol habits. The Bialystok PLUS cohort study was conducted in 2018–2022. A total of 1222 randomly selected city residents were examined and divided into two groups: before and during the COVID-19 pandemic. The participants’ lifestyle habits and medical history were collected from self-reported questionnaires. The Alcohol Use Disorders Identification Test (AUDIT) and the Fagerström Test for Nicotine Dependence (FTND) were used to assess the degree of alcohol and nicotine dependence. The survey revealed a reduced frequency of reported allergies vs. an increased frequency of reported sinusitis and asthma; increased incidence of declared hypercholesterolemia and visual impairment; a reduced number of cigarettes smoked per day, lower FTND score, and a greater desire to quit smoking in the next six months; and an increase in hs-CRP and FeNO levels in the population during the pandemic compared to the pre-pandemic population. The COVID-19 pandemic had a measurable impact on the general population’s prevalence of certain medical conditions and lifestyle habits. Further research should continue to examine the long-term health implications of the pandemic.

## 1. Introduction

Coronavirus disease 2019 (COVID-19) is a respiratory infection induced by severe acute respiratory syndrome coronavirus 2 (SARS-CoV-2), also known as coronavirus [1]. The outbreak began in China, but has spread to all countries around the world, and the number of cases outside of East Asia exceeded those in China by 15 March 2020, and rose exponentially. The number of fatalities in several countries now exceeds the total in the epidemic focus [2]. The COVID-19 pandemic remains a primary concern for global health and society. Clinicians must keep up with the data generated from across the globe, since SARS-CoV-2 emerged in December 2019 [3]. In Poland, from January 2020 to December 2022, 6,364,708 confirmed cases of COVID-19, including 118,467 deaths, were reported to the WHO [4]. The introduction of lockdown in Poland has significantly affected the nature of citizens’ work and human relations by limiting social interactions and mobility.

Health complications and lifestyle habits associated with the COVID-19 pandemic are currently being investigated. The percentage of daily tobacco smokers in 2019 in Poland was 24% among men and 18% among women [5]. Smoking is a significant risk factor for major causes of premature mortality, such as cancer and cardiovascular disease. Using cigarettes also increases the risk of heart attack, stroke, lung cancer, and cancers of the larynx and mouth. In addition, smoking is an essential contributor to respiratory diseases. Alcohol abuse also causes significant mortality and morbidity. Alcohol use disorders are usually linked to depressive episodes, severe anxiety, and insomnia. They can increase the incidence of heart disease, stroke, cancer, and cirrhosis by affecting the cardiovascular, digestive, and immune systems [6]. According to a 2019 analysis, the average level of alcohol consumption in Poland was 11 L of pure alcohol per capita (a person aged 15 years or older) [7]. 

The aim of the study was to compare the impact of pandemic COVID-19 on self-rated health status and smoking and alcohol habits. We used the Bialystok PLUS cohort, that started enrollment in 2018 and continued during pandemic, which allows for reliable data analysis before and during the pandemic. Therefore, we compared both populations—before and during the pandemic—studied using the same methodology. 

## 2. Materials and Methods

The Bialystok PLUS cohort study was conducted in 2018–2022 on a sample of Bialystok (Poland) citizens. Based on a representative sample of the residents of Bialystok city, the Bialystok PLUS study describes the health status of the adult population in northeastern Poland. The survey not only addresses the present health status of the people, thus providing valuable data on risk factors and the development of diseases, but also examines the sociological and psychological determinants that may influence them [8]. Researchers received a pseudonymized list of residents aged 20–79 from the Bialystok City Hall each year. They were then assigned categories by gender and 5-year ranges. Subsequently, samples of citizens from each subcategory were randomly selected separately, in such numbers as to obtain a distribution of proportions similar to that of the city’s population. The exact drawing procedure is described elsewhere [9]. The analyzed population, a total of 1222 respondents (570 men and 652 women), was divided into two groups: before the COVID-19 pandemic and during the COVID-19 pandemic. The beginning of the pandemic in Poland was considered to be 20 March 2020, when an epidemic state was imposed in Poland. A total of 713 probands tested between 5 November 2018 and 17 March 2020 were classified as a pre-pandemic group, and 509 probands tested between 14 July 2020 (the study had to be interrupted during lockdown between March and July) and 13 May 2022 were analyzed during the pandemic. 

The participants’ lifestyle habits and medical history were collected from self-reported questionnaires. All study participants underwent laboratory assessment, spirometry, and nitric oxide (NO) measurement in the exhaled air–fractional nitric oxide (FeNO) using nitric oxide analyzer FeNO+ from Medisoft. Anthropometric measurements included measurement of height and weight. Body mass index (BMI) was calculated as weight in kilograms divided by height in meters squared. The waist-to-hip ratio (WHR) was calculated as the circumference ratio between the waist and hips. From all patients, peripheral intravenous fasting blood samples were collected for laboratory tests, at the time of visit in the morning after eight hours of fasting. High-sensitivity C-reactive protein (hs-CRP) and a complete blood count with differential were established to assess the inflammatory state. 

The Alcohol Use Disorders Identification Test (AUDIT) was used to assess the degree of alcohol dependence. AUDIT is a practical and simple screening test for unhealthy alcohol use, defined as risky or hazardous drinking or any alcohol use disorder. Following its publication in 1989 by the World Health Organization, AUDIT has become the most widely used alcohol use screening tool in the world [10]. In addition to questions about alcohol use scored on a scale of 0 to 4, the AUDIT asks about common alcohol-related problems that patients might face, including general symptoms of alcohol dependence [11].

The Fagerström Test for Nicotine Dependence (FTND) was used to assess ordinal measures of nicotine dependence related to cigarette smoking. FTND was developed by Karl-Olov Fagerström in 1978. The instrument was modified in 1991 by Todd Heatherton et al. to form the Fagerström Test for Nicotine Dependence [12]. When assessing nicotine dependence in the Fagerström Test, yes/no questions were scored on a scale of 0 to 1, and multiple-choice questions were scored on a scale of 0 to 3. The higher the summed Fagerström score, the more intense the patient’s physical dependence on nicotine.

The Satisfaction with Life Scale (SWLS) has been used to measure a life satisfaction component of subjective well-being. SWLS was developed in 1959 by Diener [13] and has been shown to correlate with mental health measures and predict future behavior.

The questionnaire interview included an assessment of health status at the time of the study, including questions from AUDIT, FTND, and SWLS, lifestyle habits, education level, earnings, and medical history.

## 3. Statistical Analysis

The statistical analysis was performed using Stata 13.0 statistical software. Descriptive statistics for quantitative variables were presented as means and standard deviations (SD). The normality of distributions was assessed using the Shapiro–Wilk test. Values of normally distributed data were compared by unpaired t-test, whereas not-normally distributed continuous data were compared by Mann–Whitney U test. Categorical variables are displayed as frequency distributions (*n*) and simple percentages (%). The Chi^2^ test was used for the univariate comparison between the groups for categorical variables. Spearman correlation was used to analyze the dependence between the variables. The statistical significance was considered when *p* ≤ 0.05. 

## 4. Results

The analyzed populations did not differ in terms of age (*p* = 0.487), sex (*p* = 0.219), height (*p* = 0.179), weight (*p* = 0.072), or body mass index (*p* = 0.388). In contrast, WHR was statistically significantly lower in the population during the pandemic (*p* = 0.006). In the population during the COVID-19 pandemic, examination of exhaled air from the respiratory tract showed higher concentrations of FeNO (21.3 ± 18.8 ppb vs. 18.7 ± 15.4 ppb, *p* < 0.001) and higher levels of high-sensitivity C-reactive protein (1.7 ± 3.1 mg/L vs. 1.5 ± 3.6 mg/L, *p* = 0.001) compared to the pre-pandemic group. Detailed data can be found in Table 1.

A significantly lower number of reported allergies was observed in the population during the pandemic (22.8%) compared to the pre-pandemic period (23.6%; *p* = 0.042). In contrast, the incidence of reported asthma and sinusitis was significantly higher during the pandemic (6.3% and 24.6%) than before the pandemic (2.9%; *p* < 0.001 and 24.0%; *p* = 0.006). In addition, visual impairment and hypercholesterolemia were declared more frequently in the population during the COVID-19 pandemic. Detailed data can be found in Figure 1 and Table 2.

The prevalence of smoking history remained similar between the two groups. At the same time, numerically fewer individuals reported current smoking in the COVID-19 population compared to the pre-pandemic population, although without statistical significance. The average number of cigarettes smoked per day decreased during the pandemic compared with the pre-pandemic period (10.7 ± 7.1 vs. 14.0 ± 9.1; *p* = 0.005). In addition, more individuals declared planning to quit smoking in the next six months during the pandemic (39.4%) than before it (33.8%; *p* = 0.049), which was also reflected in the FTND results. In the population during the pandemic, the FTND score was statistically significantly lower than before the pandemic (2.4 ± 2.2 vs. 3.2 ± 2.3; *p* = 0.007). 

Alcohol consumption patterns were generally not different between the study groups. Only the frequency of drinking liquors, fruit liqueurs, and drinks in the past 30 days was lower during the pandemic than before the pandemic (2.3 ± 2.3 vs. 2.9 ± 2.7; *p* = 0.018), while other drinking frequencies remained unchanged. The results of the Alcohol Use Disorder Identification Test also remained consistent (*p* = 0.329) between both cohorts.

These results suggest a moderate change in smoking habits during the COVID-19 pandemic, while no differences were shown regarding alcohol drinking. 

Subjective well-being, as measured by the SWLS scale, did not statistically differ between the study groups. Detailed data can be found in Table 3.

The analysis aimed to determine the relationship between the declared intention to quit smoking and various variables, such as age, sex, BMI, education, income, and FeNO levels. Results revealed no significant associations between the intention to quit smoking and any of these variables, which may suggest that the intention to quit smoking may be independent of these factors. Detailed data can be found in Table 4.

## 5. Discussion

The current study provides data on the incidence of major diseases, as well as smoking and drinking habits in the population during the COVID-19 pandemic compared to the pre-pandemic population. The survey revealed: (1) a reduced frequency of reported allergies vs. an increased frequency of reported sinusitis, asthma; (2) an increased incidence of declared hypercholesterolemia and visual impairment; (3) a reduced number of cigarettes smoked per day, lower FTND score, and a greater desire to quit smoking in the next six months; and (4) an increase in hs-CRP and FeNO levels in the population during the pandemic compared to the pre-pandemic COVID-19 population. No differences were found in alcohol consumption habits. The current study found that the COVID-19 pandemic had a measurable impact on the general population’s prevalence of certain diseases and lifestyle habits.

Smoking is a leading factor in preventable morbidity and mortality worldwide. More than a billion people smoke, and without a significant increase in smoking cessation, at least half of them will die prematurely from tobacco-related complications [14]. Tetik B. K. et al. [15] studied patients who attended smoking cessation clinics in 2018 and were asked about their smoking cessation status after one year and after the COVID-19 pandemic. They investigated whether the pandemic had an impact on smoking cessation. When the success of those who quit smoking before the pandemic and those who quit smoking after the pandemic was compared, a statistically significant relationship was found (*p* < 0.001): the cessation rate after one year was 23.7%. In contrast, the cessation rate during the pandemic period was 31.1%. Nyman, A.L. et al. [16] conducted a study in which a sample of 1223 US adult cigarette smokers participated in an online survey in October–November 2020 to assess their perceptions of COVID-19 risk and changes in smoking, willingness to quit, and quit attempts during the COVID-19 pandemic. More smokers believed that smoking could increase COVID-19 severity than believed that smoking made them more susceptible to COVID-19. Greater perceptions of overall COVID-19 severity were associated with an increased likelihood of reduced smoking, greater readiness to quit smoking, and a greater chance of attempting to quit smoking. Gül Nur Çelik F. et al. [17] included a total of 749 respondents between the ages of 19 and 35 in the study. The degree of nicotine dependence was examined using the FTND. The mean nicotine dependence scores before the pandemic and COVID-19 were 3.03 and 2.97, respectively. A difference was observed before the pandemic (*p* = 0.002) and during the pandemic (*p* = 0.005) for health sciences students and others. Compared to before and during the pandemic, the mean addiction score was significantly lower for students whose parents were non-smokers during the pandemic. Our findings also showed a decrease in the FTND score, which can be reflected in the reduction in the average number of cigarettes smoked per day during the pandemic and the increased desire to quit smoking. This may result from increased awareness of the harmful effects of smoking on respiratory health. Smokers may also perceive increased susceptibility to and severity of COVID-19 infection, potentially increasing motivation to quit [1].

A systematic review by Roberts A. et al. [18] showed a mixed picture of alcohol use during the COVID-19 pandemic. Overall, there was a trend toward increased alcohol consumption. The percentage of people consuming alcohol during the pandemic ranged from 21.7% to 72.9% in the general population samples. Mental health-related factors were the most common correlates or triggers of increased alcohol use. Killgore W.D.S. et al. [19] evaluated whether COVID-19-related lockdown, in the form of stay-at-home orders and social isolation, was associated with changes in high-risk alcohol use. A total of 5931 people completed the AUDIT at one of six-time points between April and September 2020. Over the six-month period, hazardous alcohol use and probable dependence increased month after month for those on lockdown compared to those not on restriction. The current study found no differences in alcohol consumption between the study populations, reflected in the lack of differences in AUDIT test scores.

Based on recent literature, there is scientific and clinical evidence on the subacute and long-term implications of COVID-19, which may affect numerous organ systems [20]. The prevalence of allergic diseases was increasing globally before COVID-19. Choi, H.G. et al. [21] analyzed the frequency rate of self-reported and physician-diagnosed allergic diseases of asthma, atopic dermatitis, and allergic rhinitis in the Korean population. A total of 15,469 individuals were analyzed in the National Health and Nutrition Examination Survey dataset. There were no statistically relevant differences found between the prevalence of physician-diagnosed and present allergic diseases in 2019 and 2020 (asthma, *p* = 0.667 and *p* = 0.268; atopic dermatitis, *p* = 0.268 and *p* = 0.973; allergic rhinitis, *p* = 0.691 and *p* = 0.942, respectively). Among the Korean population, the prevalence of allergic diseases: asthma, atopic dermatitis, and allergic rhinitis did not decrease in 2019–2020. In comparison, the current study showed a decrease in the incidence of declared allergy, but increased reports of asthma and sinusitis. A particularly interesting finding is the increased awareness of hypercholesterolemia. More common contact with healthcare due to COVID-19 probably increased the number of blood labs. However, this may also be an effect of a preventive “40+” program, which has been introduced in Polish healthcare system and includes a packet of lab measurements, including cholesterol concentrations, available free of charge to all adults above the age of 40. 

Additionally, the current survey also analyzed nitric oxide (NO) and C-reactive protein (CRP). NO is exhaled in human breath and is a marker of airway inflammation [22]. Increased exhaled nitric oxide (NO) is associated with the effects of cytokines and inflammatory mediators in various lung diseases [23]. In asthma, increased FeNO reflects moderately well the inflammatory pathways induced by eosinophils in the central and/or peripheral airways. In COPD, airway/alveolar NO concentrations may be normal, and the role of FeNO monitoring is less clear and therefore less established than in asthma. In addition, concurrent cigarette smoking decreases FeNO [23]. CRP belongs to the pentraxin protein and is often used in clinical practice as a marker of infection and inflammation because its synthesis rapidly and dramatically increases after infection or tissue injury [24]. There is evidence that CRP is not only a marker of inflammation, but also that destabilized isoforms of CRP have pro-inflammatory properties [25]. Data from a cross-sectional study by Meryam Maamar et al. [26] of 121 patients three months after COVID-19 infection showed an association between post-COVID-19 syndrome (PCS) and the upper ranges of neutrophil count, neutrophil/lymphocyte ratio (NLR), fibrinogen, and CRP, suggesting an association between low-grade inflammation (LGI) and PCS [26]. Gul M. et al. [27] compared cardiac and inflammatory markers and echocardiographic parameters between COVID-19 patients (n = 126) and controls (n = 98). The mean follow-up period in the COVID-19 group was 58.39 ± 39.1 days. The value of CRP was significantly higher in the COVID-19 group than the control group. The researchers suggest that although the clinical and prognostic significance of cardiac and other inflammatory markers in the acute phase of COVID-19 is well known, the biomarkers can also be used to observe cardiac damage in the medium term after infection [27]. Results from the current study on CRP are in line with other studies from the COVID-19 pandemic period and indicate an association between the COVID-19 pandemic on elevated CRP in the general population.

Wang W. et al. [28] surveyed a total of 1733 and 1728 students in 2020 and 2019, respectively. The Vision Behavior Questionnaire, including exposure to a digital screen, was used to investigate the association between the eye parameter and eye health behavior. The percentage of respondents with myopia was 55.02% in 2020, which was higher than in 2019 (44.62%). The authors showed that increased exposure to a digital screen contributes to the myopia progression in children and adolescents during the COVID-19 pandemic. The current study also shows similar results. In the population surveyed during the COVID-19 pandemic, a higher percentage of people reported vision deterioration compared to the pre-COVID population.

The current study showed a reduced incidence of reported allergies compared to an increased incidence of reported sinusitis and asthma. In addition, the study showed a significant difference in FeNO concentrations and CRP levels in the studied populations. Increased concentrations were shown in the population during the COVID-19 pandemic which might be associated with a higher prevalence of declared asthma and sinusitis. Further research should be conducted to analyze this phenomenon more deeply.

## 6. Conclusions

The COVID-19 pandemic had a measurable impact on the general population’s prevalence of certain medical conditions and lifestyle habits. Future studies must be pursued to examine the long-term health impact of the pandemic, as well as the effectiveness of targeted interventions to mitigate these effects.

## 7. Study Limitation

This study has limitations. It is a single-center, cross-sectional study with a limited sample size of residents of a medium-sized city in the eastern part of the country. Nevertheless, the study has the advantage of analyzing the general population before and during the pandemic, carried out according to the same procedures, in the same research center, and by the same trained staff.

## Figures and Tables

**Figure 1 jcm-12-06241-f001:**
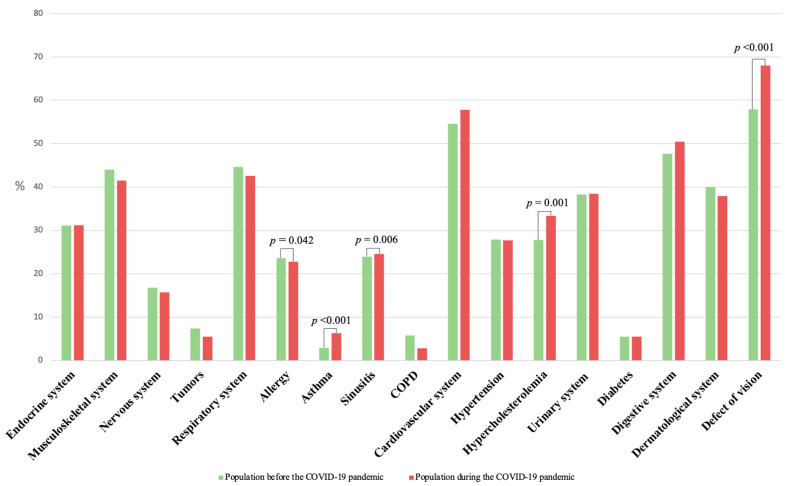
Percentage of reported diseases in the general population before and during the COVID-19 pandemic.

**Table 1 jcm-12-06241-t001:** Characteristics of the population before and during the COVID-19 pandemic; comparisons variables between subgroups.

Variable	Population before the COVID-19 Pandemic N = 713	Population during the COVID-19 Pandemic N = 509	*p*
Age, years	48.7 ± 15.4	49.3 ± 14.5	0.487
Male sex, *n* (%)	322 (45.2)	248 (48.7)	0.219
Height, cm	170.1 ± 10.1	170.8 ± 9.5	0.179
Weight, kg	77.9 ± 16.7	79.3 ± 16.2	0.072
BMI, kg/m^2^	26.8 ± 4.9	27.2 ± 5.2	0.388
WHR	0.88 ± 0.1	0.86 ± 0.1	0.006
Laboratory parameters			
hs-CRP, mg/L	1.5 ± 3.6	1.7 ± 3.1	0.001
FeNO, ppb	18.7 ± 15.4	21.3 ± 18.8	<0.001
WBC, 10^3^/uL	6.25 ± 1.6	5.8 ± 1.5	<0.001
Hgb, g/dL	14.1 ± 1.4	13.9 ± 1.3	0.057
Hct, %	41.1 ± 3.8	40.9 ± 3.7	0.396
PLT × 10^9^/L	236 ± 62.8	235.9 ± 60.3	0.821
Spirometry			
FVC, L	4.3 ± 1.2	4.3 ± 1.1	0.678
FEV1/FVC, %	79.1 ± 6.9	78.9 ± 6.6	0.302

The data are shown as *n* (%), mean ± SD. BMI: body mass index; hs-CRP: high-sensitivity C-reactive protein; WBC: white blood cells; Hct: hematocrit; Hgb: hemoglobin; PLT: platelet count; kg: kilogram; cm: centimeter; mg: milligram; L: liter; dL: deciliter; ft: femtoliter; mmHg: millimeters of mercury; SD: standard deviation; ppb: parts per billion; mg/L: milligrams per liter; mg/dL: milligrams per deciliter; FVC: forced vital capacity; FEV1: forced expiratory volume in 1 s.

**Table 2 jcm-12-06241-t002:** The incidence of reported diseases in the general population before the pandemic COVID-19 and during the pandemic COVID-19.

Disease	Population before the COVID-19 Pandemic N = 713	Population during the COVID-19 Pandemic N = 509	*p*
Endocrine system	222 (31.1)	159 (31.2)	0.901
Musculoskeletal system	314 (44.0)	211 (41.5)	0.369
Nervous system	120 (16.8)	80 (15.7)	0.636
Tumors	53 (7.4)	28 (5.5)	0.184
Respiratory system	318 (44.6)	217 (42.6)	0.678
Allergy	168 (23.6)	116 (22.8)	0.042
Asthma	21 (2.9)	32 (6.3)	<0.001
Sinusitis	171 (24.0)	125 (24.6)	0.006
COPD	41 (5.8)	14 (2.8)	0.076
Cardiovascular system	389 (54.6)	294 (57.8)	0.262
Hypertension	199 (27.9)	141 (27.7)	0.469
Hypercholesterolemia	198 (27.8)	170 (33.4)	0.001
Urinary system	273 (38.3)	196 (38.5)	0.941
Diabetes	39 (5.5)	28 (5.5)	0.168
Digestive system	340 (47.7)	257 (50.5)	0.270
Dermatological system	285 (40.0)	193 (37.9)	0.506
Defect of vision	413 (57.9)	346 (68.0)	<0.001

The data are shown as *n* (%); COPD: chronic obstructive pulmonary disease.

**Table 3 jcm-12-06241-t003:** Population behavior of the population before and during the COVID-19 pandemic.

Lifestyle Habits	Population before the COVID-19 Pandemic N = 713	Population during the COVID-19 Pandemic N = 509	*p*
Smoking habits			
Ever smoked cigarettes	402 (56.4)	289 (56.8)	0.998
Currently smoking	136 (19.1)	94 (18.5)	0.795
Number of cigarettes smoked during the day	14.0 ± 9.1	10.7 ± 7.1	0.005
The highest number of cigarettes smoked during the day	20.7 ± 10.6	16.7 ± 7.4	0.117
Individuals planning to quit smoking within 6 months	46 (33.8)	37 (39.4)	0.049
Fagerström Test for Nicotine Dependence score	3.2 ± 2.3	2.4 ± 2.2	0.007
Drinking habits			
Drinking alcoholic beverages in the last 30 days	512 (71.8)	361 (70.9)	0.564
Drinking beer in the last 30 days	326 (45.7)	195 (38.3)	0.404
Drinking alcohol in the last 30 days	232 (32.5)	147 (28.9)	0.930
Drinking liqueurs, fruit liqueurs, drink in the last 30 days	115 (16.1)	70 (13.8)	0.808
Drinking wine in the last 30 days	222 (31.1)	152 (29.9)	0.300
How many times have a beer been drunk in the last 30 days	5.9 ± 6.7	9.6 ± 5.4	0.908
Beer quantity in milliliters drunk in the last 30 days	2243.5 ± 3221.5	1327.8 ± 1337.2	0.991
How many times has alcohol been drunk in the last 30 days	2.7 ± 2.5	3.0 ± 3.7	0.720
Alcohol quantity in milliliters drunk in the last 30 days	331.9 ± 417.9	324.2 ± 364.4	0.4366
How many times has liqueurs, fruit liqueurs, drinks drunk in the last 30 days	2.9 ± 2.7	2.3 ± 2.3	0.018
Liqueurs, fruit liqueurs, drink quantity in milliliters drunk in the last 30 days	195.8 ± 282.2	339.9 ± 449.5	0.052
How many times have wine been drunk in the last 30 days	4.9 ± 4.1	2.6 ± 2.2	0.153
Wine quantity in milliliters drunk in the last 30 days	343.7 ± 427	389.4 ± 403.7	0.144
Alcohol Use Disorders Identification Test score	4 ± 3.7	3.6 ± 3.0	0.329
The Satisfaction with Life Scale (SWLS)			
SWLS	22.7 ± 5.2	22.4 ± 5.2	0.407

The data are shown as *n* (%), mean ± SD.

**Table 4 jcm-12-06241-t004:** Dependence between declared intention to quit smoking and the variables.

Variable	*p*
Age	0.724
Sex	0.327
BMI	0.184
Education	0.493
Income	0.482
FeNO, ppb	0.722

## Data Availability

The datasets are not publicly available because the individual privacy of the participants should be protected. Data are however available from the corresponding author on reasonable request.

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
