# Peer review of "The Effect of the COVID-19 Pandemic on Self-Reported Health Status and Smoking and Drinking Habits in the General Urban Population"

_jcm, 2023, doi:10.3390/jcm12196241_

Round 1

Reviewer 1 Report

The study of Chlabicz and their co-authors is interesting, the manuscript is well-written. There are a few minor comments.

1. To improve the visibility of the manuscript and make it easy for readers, I would suggest presenting some key data as a graph instead of a table format. Table 2  (incidence of reported diseases in the population before and during the pandemic) would be the best for it.

2. It is worth including a  paragraph in the discussion describing how the alcohol consumption and smoking habits of the general population have been changed due to fear/hope of new vaccines during mass vaccination against COVID-19.

Reviewer 2 Report

Thank you for inviting me to review this manuscript. The manuscript reports a study on the effect of the COVID-19 pandemic on self-reported health status, smoking, and drinking habits in the general urban population. I am afraid to say this study suffers from a fundamental methodological shortcoming. In fact, the study reports findings that are not comparable. To confirm such observations there is a need to compare the same people before and during COVID-19. We are not aware if the groups were similar in study outcomes. I think this is a fundamental problem. To claim the study conclusion parid t-test with a single sample is needed. The best way is also to have a control group. To the best of my understanding, the findings are not sound. The authors claimed 'The COVID-19 pandemic had a measurable impact on the general population's prevalence of certain medical conditions and lifestyle habits'. Then, one might argue if this is true we need to praise the COVID-19 pandemic condition since it could be useful for our health and some medical conditions!  conditions. 

Reviewer 3 Report

Were also analyzyed differences between the groups in the probands' educational attainment or their social status or their fyzical acivitys? How do you explain the change in WHR at the same BMI?

Round 2

Reviewer 2 Report

Thank you for inviting me to review this revised version. I appreciate the authors' gentle response. However, still, I am not sure if the findings are sound. Perhaps this might be due to my misunderstanding of what the authors explain.